# The Role of Mid-Trimester BUN and Creatinine Assessment in Predicting Preeclampsia: Retrospective Case–Control Study

**DOI:** 10.3390/medicina61040746

**Published:** 2025-04-18

**Authors:** Ebru Celik Kavak, Cigdem Akcabay, Meryem Demircan, Ibrahim Batmaz, Cengiz Sanli, Ahmet Senocak, Mesut Ali Haliscelik, Miray Onat, Batuhan Tepe, Salih Burcin Kavak

**Affiliations:** 1Private Practice, 32100 Elazig, Turkey; eckavak1@gmail.com; 2Department of Obstetrics and Gynecology, Firat University Faculty of Medicine, 23100 Elazig, Turkey; cagcabay@hotmail.com (C.A.); miray.sari23@hotmail.com (M.O.); batuhan_tepe_23@hotmail.com (B.T.); 3Gynecology and Obstetrics Clinic, Fethi Sekin City Hospital, 23100 Elazig, Turkey; drmrymdmr@gmail.com (M.D.); drsanlicengiz@gmail.com (C.S.); 4Department of Obstetrics and Gynecology, Faculty of Medicine, Mardin Artuklu University, 47100 Mardin, Turkey; dribrahimbatmaz@gmail.com; 5Gynecology and Obstetrics Clinic, Private East Anatolia Hospital, 23100 Elazig, Turkey; dr.ahmetsenocak@hotmail.com; 6Department of Obstetrics and Gynecology, Gazi Yaşargil Training and Research Hospital, Health Sciences University, 21500 Diyarbakir, Turkey; mesuthaliscelik@hotmail.com

**Keywords:** blood urea nitrogen, mid-trimester biomarkers, preeclampsia, prediction, serum creatinine

## Abstract

*Background and Objectives:* Preeclampsia (PE) is a major cause of adverse perinatal outcomes. Early diagnosis of pregnant women at risk of PE can facilitate disease prevention and management. However, the presence of different phenotypes of the disease complicates its prediction. In particular, the challenges in the early diagnosis of term PE cases necessitate research on PE prediction in the second and third trimesters. This study aims to examine the association between PE development and mid-trimester blood urea nitrogen (BUN), serum creatinine, and the BUN/creatinine ratio in pregnant women. *Materials and Methods:* This retrospective case–control study was conducted on women diagnosed with PE. Pregnant women who underwent routine biochemical blood tests between the 18th and 24th weeks of gestation and subsequently gave birth at our hospital between January 2022 and May 2023 were categorized into three groups. Accordingly, healthy women who had term deliveries were classified as Group 1 (150 cases), women diagnosed with PE were classified as Group 2 (58 cases), and those diagnosed with severe PE were classified as Group 3 (44 cases). *Results:* There were no significant differences in age, gravidity, parity, body mass index, or gestational week at blood sampling between the patient and control groups (*p* > 0.05). When comparing the mean blood urea nitrogen, serum creatinine, and BUN/creatinine ratios, a significant difference was observed between the control group and those who developed PE (*p* = 0.001, *p* < 0.001, and *p* = 0.031, respectively). Univariate analysis revealed a significant association between BUN levels and PE development (OR 1.083; 95% CI, 1.031–1.139; *p* = 0.002). A stronger association was observed between serum creatinine levels and PE development (OR 112.344; 95% CI, 11.649–1083.416; *p* < 0.001). However, no significant association was found between the BUN/creatinine ratio and PE in univariate analysis (OR 1.003; 95% CI, 0.979–1.028; *p* > 0.05). Mid-trimester BUN and serum creatinine levels were significantly higher in patients who developed PE and severe PE. The AUC value for the BUN parameter in predicting PE was 0.614 (AUC 0.614; 95% CI, 0.539–0.689; *p* = 0.002). A BUN cut-off value of 16.2 mg/dL predicted disease development with a sensitivity of 52.9% and a specificity of 74%. Similarly, the AUC value for the serum creatinine parameter in predicting PE was 0.644 (AUC 0.644; 95% CI, 0.574–0.751; *p* < 0.001). A serum creatinine cut-off value of 0.58 mg/dL was able to predict disease development with 37.2% sensitivity and 88% specificity. No significant AUC value was obtained for the BUN/creatinine ratio (*p* > 0.05). *Conclusions:* Our findings indicate that elevated BUN and serum creatinine levels measured during the mid-trimester (18–24 weeks) are associated with an increased risk of developing preeclampsia.

## 1. Introduction

Preeclampsia (PE) is a multisystemic disease affecting 2–8% of pregnancies and remains one of the leading causes of perinatal mortality and morbidity [1]. Gestational hypertension accounts for 14% of maternal deaths, and approximately 500,000 newborns die each year as a result of PE [2,3]. The pathophysiology of PE has not yet been fully elucidated, and there is currently no effective treatment. Therefore, early prediction and prevention remain central priorities in ongoing research.

Most predictive models for PE have focused on the first trimester. The most comprehensive of these was published in 2020 [4] and utilizes a multivariate Gaussian distribution model that integrates biochemical and biophysical parameters, including maternal characteristics, mean arterial pressure, uterine artery Doppler measurements, and placental growth factor levels. This model identified 94% of early-onset PE cases with a 10% false-positive rate (AUC 0.96, 95% CI, 0.94–0.98). Although early-onset PE, which begins before 34 weeks, is associated with more severe clinical manifestations, approximately 80% of cases present as late-onset PE after 34 weeks of gestation. Current first-trimester screening tools remain inadequate for predicting term PE [5]. As a result, there is an ongoing need to identify individuals at risk for PE in the second and third trimesters of pregnancy.

Recent literature supports the implementation of PE screening at 24, 28, 32, and 36 weeks of gestation, similar to early screening strategies [6,7]. However, the predictive power of currently used biomarkers diminishes as pregnancy advances. This decline highlights the need to identify novel biomarkers for use in the second and/or third trimesters.

Undoubtedly, the most pronounced effects of PE are observed in the kidneys, and the condition is considered the most prevalent glomerular disease worldwide [8]. Irrespective of its clinical form, the diagnostic criteria for PE include renal parameters such as proteinuria and/or elevated serum creatinine levels [9]. Proteinuria develops as a consequence of glomerular endothelial damage and increased capillary permeability [10]. While renal plasma flow and glomerular filtration rate increase during normal pregnancy, these physiological adaptations are impaired in PE, resulting in elevated serum creatinine levels [11]. Furthermore, renal microvascular dysfunction is regarded as a significant manifestation of systemic endothelial injury [12]. Accordingly, renal parameters are considered important biomarkers for both the early diagnosis and monitoring of PE.

Given the systemic nature of PE, early changes in renal function may serve as preliminary indicators of disease development. To the best of our knowledge, this is the first study to investigate mid-trimester blood urea nitrogen (BUN) and serum creatinine levels as potential early biomarkers for PE, positioning it as a pilot study to inform future research. In this study, we specifically analyzed BUN and creatinine levels measured between 18 and 24 weeks of gestation in women who subsequently developed PE, aiming to evaluate their predictive value for early identification of the syndrome.

## 2. Materials and Methods

### 2.1. Patients

This study was designed as a retrospective case–control study. The medical records of pregnant women who underwent routine biochemical blood tests between 18 and 24 weeks of gestation and subsequently gave birth at Elazig Firat University between January 2022 and May 2023 were retrospectively reviewed. Prior to data collection, the study received approval from the local ethics committee (Ethics Committee Approval Number: 2022/11-02) and permission from the provincial health directorate to access patient data. The control group consisted of healthy, normotensive women who delivered at term (≥37 weeks) and did not develop obstetric complications during pregnancy. These women were matched with those diagnosed with PE and severe PE based on age, gravidity, parity, and body mass index. Accordingly, the study groups were categorized as follows: Group 1 (150 cases) included healthy women who delivered at term, Group 2 (58 cases) consisted of women diagnosed with PE, and Group 3 (44 cases) comprised women diagnosed with severe PE.

In 2013, the American College of Obstetricians and Gynecologists (ACOG) revised the diagnostic criteria for PE, with the latest update released in June 2020, finalizing the new classification principles [9]. The diagnostic criteria outlined in this guideline were used to identify patients with PE and severe PE.

According to these criteria, PE was diagnosed in women with previously normal blood pressure if, after 20 weeks of gestation, systolic blood pressure exceeded 140 mmHg or diastolic blood pressure exceeded 90 mmHg on at least two occasions at least four hours apart, with or without proteinuria, accompanied by at least one systemic finding.

For severe PE diagnosis, the presence of systolic blood pressure >160 mmHg or diastolic blood pressure >110 mmHg, regardless of proteinuria, along with at least one systemic finding, was required.

The proteinuria criteria used for diagnosis included at least one of the following: ≥300 mg proteinuria in a 24 h urine sample, a protein/creatinine ratio >0.3, or ≥+1 proteinuria detected by dipstick testing in a spot urine sample. Systemic findings included thrombocytopenia (platelet count <100,000/mm^3^), renal insufficiency (serum creatinine >1.1 mg/dL, or a baseline creatinine level at least twice the normal value in the absence of other renal diseases), hepatic dysfunction (elevated liver transaminases at least two times the upper normal limit), pulmonary edema, or the presence of cerebral or visual symptoms.

Women with multiple pregnancies, chronic or gestational hypertension, obesity (BMI > 30 kg/m^2^), advanced maternal age (>35 years), smoking, kidney disease, diabetes mellitus, thrombophilia, or connective tissue disorders were excluded from the study. Additionally, patients using medication for any disease were also excluded from the study. Maternal demographic characteristics and other medical data were obtained from medical records.

Laboratory data included renal function parameters, specifically blood urea nitrogen (BUN), serum creatinine, and the BUN/creatinine ratio measured between 18 and 24 weeks of gestation.

### 2.2. Statistical Analysis

Data were analyzed using IBM Statistical Package for the Social Sciences (SPSS) version 23. The normality of data distribution was assessed using the Shapiro–Wilk and Kolmogorov–Smirnov tests. As the majority of variables did not follow a normal distribution, non-parametric tests were used for group comparisons. The Chi-square test was used to compare categorical variables between groups. For comparisons of non-normally distributed data among three or more groups, the Kruskal–Wallis test was employed, followed by the Dunn test for multiple comparisons.

Binary logistic regression analysis was performed to evaluate risk factors associated with preeclampsia, and the results were reported as odds ratios (ORs) with 95% confidence intervals (CIs). The sensitivity and specificity of different threshold values for each variable (BUN, serum creatinine, BUN/creatinine ratio) in detecting PE were calculated and presented as receiver operating characteristic (ROC) curves.

Multivariable logistic regression analysis was applied to identify independent predictors of PE and severe PE, with results reported as odds ratios (ORs) and 95% confidence intervals (CIs). Continuous variables were expressed as mean ± standard deviation, and categorical variables were presented as counts (percentages). A *p*-value of <0.05 was considered statistically significant.

## 3. Results

Between January 2022 and May 2023, a total of 1331 deliveries were recorded at Elazig Firat University. During this period, 119 patients were diagnosed with PE (8.9%), and 57 patients were diagnosed with severe PE (4.2%), as determined from hospital records. Among these, 58 pregnant women with preeclampsia who had available mid-trimester biochemical measurements were classified as Group 2, while 44 women diagnosed with severe PE were categorized as Group 3. The control group (Group 1) consisted of 150 healthy, term-delivering pregnant women who were matched and had accessible mid-trimester laboratory data.

The gravidity and parity characteristics of the patients are presented in Table 1. There was no significant difference in gravidity and parity between the groups (*p* > 0.05). In Group 1, 72% of cases had a gravidity of 1, compared to 67.2% in Group 2, and 75% in Group 3 (*p* = 0.851). Regarding parity, 82.7% of cases in Group 1, 77.2% in Group 2, and 79.5% in Group 3 had a parity of 0 (*p* = 0.907).

A comparison of the obstetric and demographic data, as well as renal parameters of the patients, is presented in Table 2.

A significant difference in the gestational week at delivery was observed between Groups 1 and 3 (*p* = 0.002) and between Groups 2 and 3 (*p* = 0.004). Median BUN levels were significantly higher in Group 2 compared to Group 1 (*p* = 0.001). Serum creatinine levels also showed significant differences, with both Group 2 and Group 3 having higher values than Group 1 (*p* < 0.001). Additionally, a significant difference in the BUN/creatinine ratio was identified between Group 2 and Group 3 (*p* = 0.031).

To evaluate mid-trimester BUN, serum creatinine, and BUN/creatinine values as potential risk factors for PE, binary logistic regression analysis was performed using both univariate and multivariate models. The univariate model showed that a one-unit increase in BUN levels is associated with a 1.08-fold increase in the risk of PE (*p* = 0.002). A rise in serum creatinine levels was associated with a 112.34-fold increased risk of PE (*p* < 0.001). In multivariate model 1 (excluding the BUN/creatinine ratio), a one-unit increase in serum creatinine levels was found to increase the risk of PE by 59.74 times (*p* = 0.001). In multivariate model 2, which included the BUN/creatinine ratio, no significant effect of any variable was found (*p* > 0.050). The odds ratio values with 95% confidence intervals for laboratory parameters are presented in Table 3.

For PE prediction, the area under the curve (AUC) value for BUN was 0.614, which was statistically significant (*p* = 0.002). Based on this, a BUN cut-off value of 16.2 mg/dL yielded a sensitivity of 52.94%, specificity of 74%, positive predictive value (PPV) of 58.06%, and negative predictive value (NPV) of 69.81%. The AUC value for serum creatinine was 0.644, which was also statistically significant (*p* < 0.001). With a serum creatinine cut-off value of 0.58 mg/dL, the sensitivity, specificity, PPV, and NPV were determined to be 37.25%, 88%, 67.86%, and 67.35%, respectively. No significant AUC value was obtained for the BUN/creatinine ratio (*p* = 0.936).

The ROC analysis results for BUN, serum creatinine, and BUN/creatinine are presented in Table 4.

A boxplot graph for serum creatinine values is shown in Figure 1.

The receiver operating characteristic (ROC) curve for BUN, serum creatinine, and BUN/creatinine values is shown in Figure 2.

## 4. Discussion

In this study, we found that blood urea nitrogen (BUN) and serum creatinine levels measured during the mid-trimester (18–24 weeks) were significantly elevated in women who subsequently developed preeclampsia. Furthermore, threshold values were established for both parameters: BUN levels > 16.2 mg/dL and serum creatinine >0.58 mg/dL were significantly associated with an increased risk of PE. In contrast, the BUN/creatinine ratio did not exhibit statistically significant predictive value for PE development.

The etiology and pathogenesis of PE remain incompletely elucidated. The condition comprises a heterogeneous spectrum of clinical subtypes that emerge from a complex interplay of genetic, immunological, and environmental factors [13]. Clinically, PE may present in mild or severe forms, as early-onset (<34 weeks) or late-onset (>34 weeks), and is often accompanied by fetal growth restriction, HELLP syndrome, or chronic hypertension. This phenotypic diversity reinforces the notion that PE is more appropriately considered a syndrome rather than a singular disease entity, thereby complicating its prediction, classification, and clinical management [14].

The “two-stage model” remains the most widely accepted pathogenic framework for preeclampsia (PE). In the first stage, inadequate remodeling of the spiral arteries during early pregnancy leads to impaired placental perfusion and ischemia. The second stage is characterized by systemic endothelial dysfunction, accompanied by inflammation and oxidative stress, which ultimately result in the clinical manifestations of PE [12,15]. Spiral artery invasion is typically completed between 8 and 22 weeks of gestation, and disruption of this physiological process is considered a pivotal event in the pathogenesis of the disorder [16,17]. A variety of molecular mediators—including cytokines, growth factors, placental growth factor (PlGF), vascular endothelial growth factor (VEGF), and soluble fms-like tyrosine kinase-1 (sFlt-1)—have been implicated in these processes and are actively studied as predictive biomarkers for PE [18].

Currently, first-trimester combined screening—incorporating maternal risk factors, mean arterial pressure (MAP), uterine artery pulsatility index (UtA-PI), and placental growth factor (PlGF)—is regarded as one of the most effective strategies for predicting preeclampsia (PE). In individuals classified as high-risk using this approach, initiating low-dose aspirin therapy before 16 weeks of gestation has been shown to reduce the risk of preterm PE by up to 62% [19]. However, its sensitivity for predicting term PE remains suboptimal, with detection rates of approximately 41% [14,19].

While second-trimester studies employing similar parameters have demonstrated high predictive accuracy for early-onset preeclampsia (PE), their ability to detect term PE remains limited [7,20]. In the third trimester, both the sFlt-1/PlGF ratio—particularly when exceeding a threshold of 40—and individualized risk models have demonstrated robust performance in forecasting the short-term onset of PE [21,22].

To date, numerous biomarkers have been evaluated for the prediction of preeclampsia (PE), as no single marker has proven sufficient. This limitation reflects the multifactorial and still incompletely understood pathophysiology of the condition. For instance, models based solely on maternal risk factors may demonstrate high detection rates (89–93%); however, they are associated with false-positive rates as high as 64% [23]. Similarly, models relying exclusively on the uterine artery pulsatility index can detect early-onset PE with 77% sensitivity, but this decreases substantially to 27% for late-onset PE [24]. Consequently, integrated or “combined models” that incorporate multiple biomarkers have been increasingly proposed. The findings of our study may provide a complementary contribution to such multi-marker prediction algorithms, pending validation in larger, prospective cohorts. In recent years, growing interest has focused on the potential utility of renal biomarkers in predicting PE. Parameters such as serum creatinine, blood urea nitrogen (BUN), uric acid, proteinuria, and the protein/creatinine ratio have been investigated both for diagnostic purposes and for assessing disease severity [8,10,25,26]. Among these, elevated uric acid levels prior to clinical onset and their association with adverse maternal–fetal outcomes have positioned uric acid as a potentially valuable biomarker [27,28]. Proteinuria remains a cornerstone in PE diagnosis, with thresholds of ≥300 mg/24 h urine or a protein/creatinine ratio >0.3 widely accepted in clinical guidelines [9,26].

During normal pregnancy, physiological adaptations in renal function are well established. Renal plasma flow increases by approximately 80%, glomerular filtration rate (GFR) increases by 50%, and serum creatinine levels typically decline to around 0.4 mg/dL [11,29]. A serum creatinine level ≥1 mg/dL is generally considered indicative of renal dysfunction during pregnancy [10]. In our study, mid-trimester serum creatinine ≥0.58 mg/dL and BUN ≥16.2 mg/dL were significantly associated with an increased risk of developing PE. The AUC values for creatinine and BUN were 0.644 (*p* < 0.001) and 0.614 (*p* = 0.002), respectively. Creatinine demonstrated high specificity (88%) but limited sensitivity (37.2%). Univariate analysis revealed a statistically significant association between elevated BUN levels and PE (OR 1.083; 95% CI, 1.031–1.139; *p* = 0.002), with an even stronger association observed for serum creatinine (OR 112.344; 95% CI, 11.649–1083.416; *p* < 0.001). In contrast, the BUN/creatinine ratio did not demonstrate a significant association with PE (OR 1.003; 95% CI, 0.979–1.028; *p* > 0.05). Both BUN and creatinine levels were also significantly higher in patients who developed PE and severe PE.

Although serum creatinine demonstrated the strongest association with PE risk (OR = 112.344), we performed a multivariate regression analysis to account for potential confounding factors. In the adjusted model, the odds ratio for creatinine decreased to 59.748 but remained statistically significant (*p* = 0.001), further supporting an independent relationship between elevated creatinine levels and the development of PE. While the AUC values for BUN (0.614) and creatinine (0.644) were modest, both reached statistical significance and may hold clinical value—particularly given their wide availability, affordability, and ease of measurement. The diminished predictive performance of the multivariate model, including the BUN/creatinine ratio, likely reflects collinearity between the underlying variables. These findings suggest that although renal biomarkers alone may not be sufficient for comprehensive PE screening, they could serve as accessible and cost-effective adjuncts when incorporated into broader, multi-marker predictive models.

Given the heterogeneous nature of PE, achieving high predictive accuracy with only a few biomarkers remains challenging. Nonetheless, incorporating BUN and creatinine into future predictive models may improve detection rates. Notably, these biomarkers are inexpensive, widely available, and do not require specialized assays, offering practical and economical advantages—particularly in resource-limited healthcare settings.

This study possesses several methodological strengths. All clinical and laboratory procedures were carried out at a single center using standardized protocols, ensuring consistency in data collection. PE diagnoses were made by the same research team in accordance with current clinical guidelines, and all laboratory analyses were performed using Ministry of Health–approved systems. Importantly, one of the study’s notable strengths is the evaluation of widely available, low-cost biomarkers, such as BUN and creatinine, which enhances the clinical applicability of the findings.

Despite its strengths, this study has several limitations. The retrospective design and relatively small sample size within PE subgroups may have limited the statistical power. Additionally, the single-center nature of the study, conducted within a single geographic region, restricts the generalizability of the findings. PE cases could not be stratified by gestational age, further limiting subgroup analysis. Therefore, this study should be considered a preliminary, hypothesis-generating investigation, and its results interpreted with appropriate caution. Future large-scale, multicenter studies are warranted to validate these findings. Such studies should encompass diverse phenotypes of pregnancy-related hypertensive disorders (e.g., chronic hypertension, gestational hypertension), relevant comorbidities such as obesity and diabetes, and populations representing varied racial and socioeconomic backgrounds to enhance external validity and clinical applicability.

## 5. Conclusions

This study demonstrated that mid-trimester blood urea nitrogen (BUN) and serum creatinine levels are elevated in pregnancies that subsequently develop preeclampsia. These findings highlight the potential value of renal biomarkers in the early identification of at-risk patients. Future research should focus on integrating these parameters into multimodal predictive models, conducting prospective validations, and stratifying outcomes by preeclampsia phenotypes to improve the clinical utility and generalizability of these biomarkers.

## Figures and Tables

**Figure 1 medicina-61-00746-f001:**
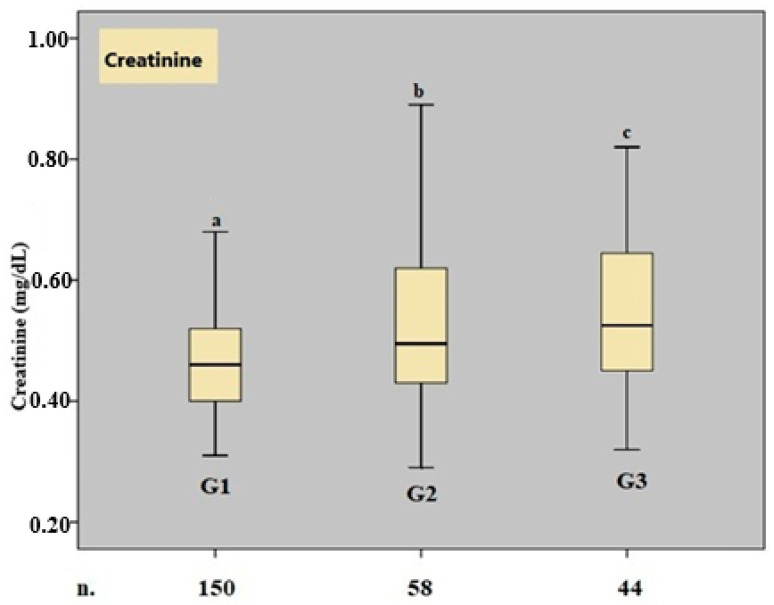
Boxplot graph of the creatinine parameter. ***n*.** number, ***G*.** group, ***a*, *b,*** and ***c*** show that there is a significant difference between the median values, *p* < 0.05, Kruskal–Wallis variance analysis.

**Figure 2 medicina-61-00746-f002:**
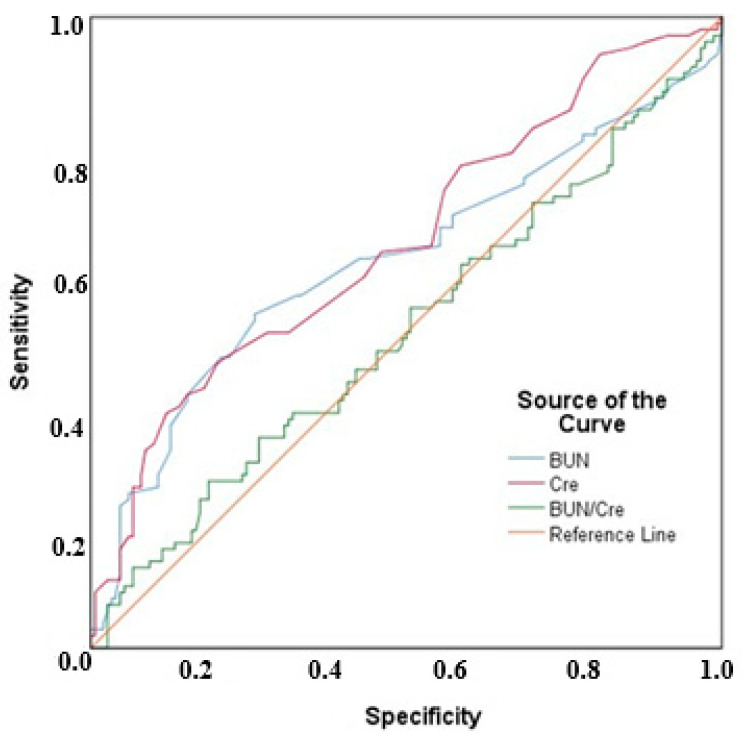
ROC curves for BUN, serum creatinine, and BUN/creatinine values.

**Table 1 medicina-61-00746-t001:** Comparison of Gravidity and Parity Among Groups.

*Parameters*	*Group 1 (%)*	*Group 2 (%)*	*Group 3 (%)*	*Total (%)*	*p-Value* ***
**Gravida (number)**					
1	108 (72)	39 (67.2)	33 (75)	180 (71.4)	0.851
2	27 (18)	13 (22.4)	6 (13.6)	46 (18.3)	
3	15 (10)	6 (10.3)	5 (11.4)	26 (10.3)	
**Parity (number)**					
0	124 (82.7)	44 (77.2)	35 (79.5)	203 (80.9)	0.907
1	23 (15.3)	11 (19.3)	8 (18.2)	42 (16.7)	
2	3 (2)	2 (3.5)	1 (2.3)	6 (2.4)	

*: Chi-square test, frequency (%).

**Table 2 medicina-61-00746-t002:** Comparison of Obstetric and Demographic Data Along with Renal Parameters Among Groups.

*Parameters*	*Group 1*	*Group 2*	*Group 3*	*p-Value **
**GW (weeks)**	20.91 ± 1.84	20.84 ± 1.83	20.77 ± 1.52	0.959
	21.00 (18.00–24.00)	21.00 (18.00–24.00)	21.00 (18.00–24.00)	
**BMI (kg/m^2^)**	24.08 ± 1.14	24.15 ± 0.97	24.23 ± 0.99	0.739
	24.10 (21.60–26.70)	24.15 (21.80–26.70)	24.15 (22.40–26.70)	
**Age (years)**	27.05 ± 4.47	25.98 ± 4.17	26.45 ± 5.21	0.299
	26.00 (18.00–34.00)	27.00 (19.00–32.00)	26.00 (18.00–34.00)	
**Birth Week (weeks)**	38.20 ± 1.30	37.50 ± 1.51	31.33 ± 2.95	**0.002**
	38.00 (38.00–40.00)	37.00 (36.00–40.00)	31.00 (28.00–37.00)	
**BUN (mg/dL)**	14.91 ± 4.47	18.23 ± 7.00	16.01 ± 5.95	**0.001**
	14.00 (8.00–34.32) ***a***	18.00 (5.00–42.00) ***b***	15.00 (8.00–41.00) ***ab***	
**Crea. (mg/dL)**	0.47 ± 0.10	0.53 ± 0.14	0.55 ± 0.15	**<0.001**
	0.46 (0.31–0.95) ***a***	0.50 (0.29–0.89) ***b***	0.53 (0.32–1.00) ***b***	
**BUN/Crea.** **ratio**	32.26 ± 9.69	34.87 ± 11.89	29.52 ± 9.21	**0.031**
	30.92 (14.00–72.22) ***ab***	32.75 (7.58–61.54) ***a***	28.87 (9.76–60.29) ***b***	

*: Kruskal–Wallis test; **GW:** Gestational week at the time of blood sampling, **BMI:** body mass index, **BUN:** blood urea nitrogen, **Crea.:** serum creatinine. Groups sharing the same letters indicate no significant difference; data are presented as mean ± standard deviation, with median (minimum–maximum) values given in the lower row.

**Table 3 medicina-61-00746-t003:** Analysis of Risk Factors Affecting Preeclampsia.

	Univariate		Multivariate 1		Multivariate 2	
	OR (%95 CI)	*p-Value*	OR (%95 CI)	*p-Value*	OR (%95 CI)	*p-Value*
**BUN (mg/dL)**	1.083		1.05			
	(1.031–1.139)	**0.002**	(0.996–1.108)	0.069	---	---
**Crea. (mg/dL)**	112.344		59.748			
	(11.649–1083.416)	**<0.001**	(5.214–684.706)	**0.001**	**---**	---
**BUN/Crea.** **ratio**	1.003				1.002	
	(0.979–1.028)	0.818	---	---	(0.978–1.028)	0.845

**BUN:** blood urea nitrogen, **Crea.:** serum creatinine, **OR:** odds ratio, **CI:** confidence interval.

**Table 4 medicina-61-00746-t004:** ROC Analysis Results of Renal Function Parameters.

Parameters	Cut-Off Value	AUC (%95 CI)	*p-Value*	Sensitivity (%)	Specificity (%)	PPV (%)	NPV (%)
**BUN** (**mg/dL)**	≥16.2	0.614 (0.539–0.689)	**0.002**	52.94	74	58.06	69.81
**Crea.** (**mg/dL)**	≥0.58	0.644 (0.574–0.715)	**<0.001**	37.25	88	67.86	67.35
**BUN/Crea.** **ratio**	---	0.503 (0.429–0.577)	0.936	---	---	---	---

**BUN:** blood urea nitrogen, **Crea.:** serum creatinine, **AUC:** area under the curve, **PPV:** positive predictive value, **NPV:** negative predictive value.

## Data Availability

Datasets generated or analyzed in this study are accessible upon reasonable request from the corresponding author.

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
