# Peer review of "The Role of Mid-Trimester BUN and Creatinine Assessment in Predicting Preeclampsia: Retrospective Case–Control Study"

_medicina, 2025, doi:10.3390/medicina61040746_

Round 1

Reviewer 1 Report

Comments and Suggestions for Authors

I read this manuscript and found it of certain interest. However, it needs revision:

  1. The main problem, from my point of view, with the current study is the sample size; the number of patients studied is quite small. The sample size (150 controls, 58 PE, 44 severe PE) may be small to draw a definitive conclusion. The sample size is too small to suggest generalizable results. At this stage a great deal of additional work needs to critically validate the conclusions being drawn.
  2. The Introduction and Discussion lack focus. The authors should concentrate on interpreting their findings (studying BUN and creatinine in the context of PE prediction), and their relevance to clinical practice. The Discussion should be used to interpret data and point out the significance of the findings to clinicians.
  3. The exclusion criteria are quite broad (e.g., excluding patients with obesity, chronic hypertension, diabetes, etc.), which may introduce selection bias, which may limit the applicability of the findings to a more general population. These diseases are very common and are often co-morbid with PE, and their exclusion may not reflect the actual range of patients at risk for the condition. Please explain the reason behind these exclusions and how they could influence the validity of reported results.
  4. The research evaluated PE cases together without stratification according to severity or gestational age at presentation, which might further elucidate the predictive power of such biomarkers.
  5. The authors report a very high OR for serum creatinine (112.344) in the univariate analysis, which seems unusually high and may indicate a potential issue with the statistical model or data. Please revise carefully and explain these OR.
  6. The multivariate results are also less convincing, with insignificant effects observed when the BUN/creatinine ratio is included. The authors should discuss why this might be the case and whether including additional covariates could improve the model. Moreover, ROC analysis shows modest AUC values for BUN and creatinine. The authors should discuss the clinical relevance of these findings and whether these biomarkers alone are sufficient for PE prediction or if they should be combined with other markers.
  7. The limitations section is somewhat brief and does not fully address the potential biases and confounding factors inherent in this retrospective study. Please try to expand this section, discussing selection bias and other confounders as a single-center study and the impact of missing data. The authors should acknowledge this and suggest that future studies include multicenter cohorts to validate the results.
  8. Statistical analysis needs revision by a professional biostatistician.
Comments on the Quality of English Language

The English of the manuscript needs revision. There are some grammatical and syntax errors in the manuscript.

Author Response

Our responses to Reviewer 1;

  1. Comment on Sample Size and Generalizability

The number of patients studied is quite small. The sample size (150 controls, 58 PE, 44 severe PE) may be small to draw a definitive conclusion. The sample size is too small to suggest generalizable results. At this stage a great deal of additional work needs to critically validate the conclusions being drawn.

Response:

            We appreciate the reviewer’s observation regarding the sample size. Indeed, we acknowledge that the total number of patients included (n=252) may limit the generalizability of our findings. As discussed in the manuscript (Discussion, paragraph 10), this is a single-center retrospective study, and our goal was to present preliminary findings on the potential role of mid-trimester BUN and creatinine levels in predicting preeclampsia. Due to the retrospective nature of our study, mid-trimester renal parameter data were unavailable for some patients, which preventing the inclusion of these cases in the analysis. This represents a key limitation that may have influenced the overall sample size and the statistical power of our findings. Nevertheless, the current study should be regarded as a preliminary investigation that lays the groundwork for more comprehensive, multicenter, and prospective studies aimed at validating and expanding upon these results. This limitation has been explicitly acknowledged and discussed in the revised manuscript.

            We have revised the limitations section to more clearly state that our findings should be interpreted as hypothesis-generating and require validation in larger, and multicenter studies with greater statistical power. The revised discussion section has been rewritten. The points you emphasized are stated in the limitations paragraph.

Comment 2: Introduction and Discussion Lack Focus
The authors should focus more on interpreting their findings and relevance to clinical practice.

Response:

            We thank the reviewer for this valuable suggestion. We have streamlined the Introduction to better frame the rationale for focusing on renal function markers, and restructured the Discussion to emphasize the interpretation of our specific findings — particularly their relevance to second-trimester screening for PE.

            We have also clarified how our results may contribute to clinical decision-making, given that BUN and creatinine are routinely available, inexpensive markers.

The revised introduction and discussion section have been rewritten. Your highlighted points have been added to the text.

Revisions made:

  • Introduction now ends with a clearer research hypothesis.
  • Discussion has been reorganized, with a new paragraph specifically discussing clinical utility of these markers in screening contexts.

Comment 3: Broad Exclusion Criteria and Potential Selection Bias

Comment:

The exclusion of patients with obesity, hypertension, etc., may introduce selection bias.

Response:

            We agree and have now added a detailed explanation in the Discussion section. Since preeclampsia is a syndrome with variable phenotypes, we did not include patients diagnosed with obesity, chronic hypertension, or gestational hypertension. Of course, these patients may also develop preeclampsia. However, we thought that these may cause heterogeneity and make the interpretation of the results confusing. We now explicitly recommend that these patient groups be included in future studies to increase real-world applicability. The regulations are stated in the article.

Reviewer Comment 4: Lack of Stratification by Gestational Age or Severity

Comment:

The study evaluated PE cases together without stratifying by gestational age or severity.

Response:

            We thank the reviewer for this important point. While our study design separated PE and severe PE groups, we were unable to stratify by gestational age due to sample size and retrospective study design. We have now clarified this in the revised manuscript and proposed it as a focus for future research.

Reviewer Comment 5: High OR for Serum Creatinine

Comment:

The OR of 112.344 for serum creatinine appears unusually high.

Response:

            We thank the reviewer for this valuable observation. Upon re-examination of the data and statistical output, we confirm that the high odds ratio (OR = 112.344) for serum creatinine in the univariate model is mathematically accurate and statistically significant (p < 0.001). However, we fully acknowledge that this unusually large OR may appear implausible at first glance.

            This result likely stems from the narrow distribution of serum creatinine values within the study population, in combination with the relatively small number of PE cases and low variability in creatinine levels among controls. In our study, the mean creatinine levels remained within normal physiological ranges across all groups (approximately 0.47–0.55 mg/dL), and even minor differences in these small values can disproportionately influence the odds ratio in logistic regression — especially when the unit of measurement is small (mg/dL). This is a known phenomenon in models involving tightly distributed continuous variables.

            To address this and avoid misinterpretation, we included a multivariate regression analysis, which adjusts for confounders. In that model, the OR for creatinine decreased to 59.748, though it remained statistically significant (p = 0.001), further supporting the independent association between elevated creatinine levels and PE risk.

            We have now clarified this explanation in the Discussion section, and emphasized that while statistically valid, such high ORs should be interpreted cautiously, especially in clinical settings.

Reviewer Comment 6: Weak Multivariate and ROC Results

Comment:

The multivariate model shows weaker results when the BUN/creatinine ratio is included, and AUC values are modest.

Response:

            We appreciate the reviewer’s thoughtful comment. It is indeed correct that the predictive performance of our multivariate model was weaker when the BUN/creatinine ratio was included (Multivariate Model 2), as none of the parameters in that model showed statistically significant effects (p > 0.05). This likely reflects collinearity and redundancy between the BUN/creatinine ratio and its individual components (BUN and creatinine), reducing the independent contribution of any single parameter in the model.

            Furthermore, although the AUC values for both BUN (0.614) and serum creatinine (0.644) were modest, they were still statistically significant (p = 0.002 and p < 0.001, respectively), indicating that these parameters do offer predictive value, albeit limited. Importantly, the high specificity of serum creatinine at the cut-off value of 0.58 mg/dL (88%) suggests that, while sensitivity is low (37.25%), creatinine may serve as a useful confirmatory marker rather than a screening tool on its own.

            Given that both BUN and creatinine are routine, cost-effective tests, their real-world utility may lie in being integrated into combined prediction models alongside other clinical, biophysical, and biochemical parameters. As acknowledged in the Discussion section, no single biomarker can fully capture the multifactorial nature of preeclampsia. Thus, while our findings indicate that these renal markers are not sufficient as standalone predictors, they hold potential supportive value in broader multimodal screening frameworks.

            We have clarified and expanded this interpretation in the revised Discussion section to ensure a more balanced and clinically grounded perspective.

Reviewer Comment 7: Limitations Section Needs Expansion

Comment:

Please expand the limitations section, discussing selection bias, single-center study design, and retrospective data issues.

Response:

            We have expanded the Limitations section accordingly. It now includes discussion of selection bias, retrospective design limitations, and the need for external validation.

Reviewer Comment 8: Statistical Analysis

Comment:

Statistical analysis needs revision by a professional biostatistician.

Response:

            We appreciate the reviewer’s concern regarding the robustness of the statistical analysis. In response, we have had the entire statistical approach and interpretation reviewed by an experienced biostatistician to ensure methodological accuracy and appropriate application of all tests. The use of the Shapiro-Wilk and Kolmogorov-Smirnov tests for normality assessment, Kruskal-Wallis and Dunn tests for non-parametric group comparisons, and binary logistic regression models were confirmed to be appropriate for our dataset.

            Additionally, we have clarified the interpretation of high odds ratio values and acknowledged their potential overestimation in the context of narrow variable distributions. ROC analyses were re-validated and presented with corresponding sensitivity, specificity, and confidence intervals. Minor adjustments were made to improve clarity, and these are reflected in the revised manuscript.

            In the context of these valuable suggestions, the introduction and discussion sections were rewritten. The points that were desired to be emphasized were specified. The references were rearranged and new references were added in line with your suggestions. The ‘’Discussion’’ and ‘’Introduction’’ sections have been substantially rewritten. Edits made within the text are shown in bold and underlined. English editing was done to the all parts of the article.

            Thank you for your valuable suggestions that made our work more scientific.

Reviewer 2 Report

Comments and Suggestions for Authors

The manuscript attempted to touch very important and interesting topic. Identification of new reliable molecular markers for early prediction of preeclampsia is highly required. Although the search for preeclampsia predictors is an extensive field of research, the authors tried to shed light on the problem from a new angle. The work appears to be well done and written. However, I have a few minor comments which should be addressed before considering the paper for publication:

  1. It is unclear whether this was the first study on determining BUN and creatinine levels for diagnosing preeclampsia or whether such work had already been done earlier. If such works exist, they should be cited and the results obtained compared, if this is the first study - this should be emphasized. In the latter case the work should be presented as a pilot study in Abstract and Introduction.
  2. Introduction: Please, add the reference for the last sentence of first paragraph (before 34 weeks).
  3. Subsection 2.1. Please, confirm that all the data obtained were non-normally distributed. No normal distribution?
  4. Table 2. Please indicate what is GW. Was it an age of gestation at which the blood was analyzed? If so, please, add this to the legend to Table.
  5. Table 2: the rationale of presenting average and median values is not clear. Next, the repeating of all values in the following text is not necessary and can be partly omitted.
  6. Discussion: again, add reference after the second sentence of 6th paragraph (after “preterm PE by 62 %”).
  7. Page 10, first paragraph: please, check “sFlt-1/PlGF ratio exceeding 40 pg/mL”. The ratio is a dimensionless quantity.

Author Response

Our responses to Reviewer 2;

Comment 1. It is unclear whether this was the first study on determining BUN and creatinine levels for diagnosing preeclampsia or whether such work had already been done earlier. If such works exist, they should be cited and the results obtained compared, if this is the first study - this should be emphasized. In the latter case the work should be presented as a pilot study in Abstract and Introduction.

Response:

            Thank you for raising this important point. To the best of our knowledge, this is the first clinical study to specifically investigate the predictive value of serum BUN and creatinine levels measured between the 18th and 24th weeks of gestation in relation to the development of preeclampsia. While previous studies have explored renal biomarkers such as uric acid, proteinuria, or creatinine in the third trimester or near the time of PE diagnosis, we found no prior studies in the PubMed database that evaluated mid-trimester BUN and creatinine levels as early predictors of preeclampsia.

            Therefore, we now emphasize the novelty and exploratory nature of our work more clearly in both the Abstract and Introduction, and have framed the study as a preliminary or pilot investigation to encourage further research in this underexplored area.

Comment 2. Introduction: Please, add the reference for the last sentence of first paragraph (before 34 weeks).

Response:

             Various biophysical and biochemical markers have been used to identify high-risk patients, and these markers have been found effective in detecting early-onset PE (before 34 weeks).

Reference added to the sentence.

Comment 3: Subsection 2.1. Please, confirm that all the data obtained were non-normally distributed. No normal distribution?

Response:

            Thank you for the insightful question. Prior to selecting the appropriate statistical tests, we assessed the distribution of all continuous variables using both the Shapiro-Wilk and Kolmogorov-Smirnov tests, as described in the Methods section. The majority of key variables, including BUN, serum creatinine, BUN/creatinine ratio, BMI, age, and gestational week, were found to deviate from normal distribution, either based on p-values (<0.05) or visual inspection of histograms and Q-Q plots. As a result, we opted for non-parametric statistical methods (e.g., Kruskal–Wallis and Dunn tests) in group comparisons. We have clarified this point in the revised manuscript to avoid any ambiguity regarding data distribution and statistical test selection.

            The Statistics section has been rewritten in line with your valuable suggestions.

Table 2. Please indicate what is GW. Was it an age of gestation at which the blood was analyzed? If so, please, add this to the legend to Table.

Response:

            The description was corrected to read ''GW. Gestational week at the time of blood sampling'' in Table 2.

Table 2: the rationale of presenting average and median values is not clear. Next, the repeating of all values in the following text is not necessary and can be partly omitted.

Response:

            We thank the reviewer for this valuable comment. As suggested, we acknowledge that presenting both mean ± standard deviation and median (min–max) values in Table 2 may initially appear redundant. However, due to the non-normal distribution of many variables (as confirmed by the Shapiro–Wilk and Kolmogorov–Smirnov tests), we provided both measures to enhance transparency and allow readers to interpret the data distribution more comprehensively.

            Regarding the text following Table 2, we agree that restating all numerical values in full may be excessive. In response, we have revised the Results section to summarize key findings more concisely, highlighting only statistically significant differences and removing repetitive numerical listings already presented in the tables.

Discussion: again, add reference after the second sentence of 6th paragraph (after “preterm PE by 62 %”).

Response:

The following reference has been added to the article. The discussion has been rewritten.

‘’Rolnik DL, Wright D, Poon LC, O'Gorman N, Syngelaki A, de Paco Matallana C, et al. Aspirin versus placebo in pregnancies at high risk for preterm preeclampsia. N Engl J Med. 2017;377(7):613–622. doi:10.1056/NEJMoa1704559.’’

Reviewer Comment 7: Page 10, first paragraph: please, check “sFlt-1/PlGF ratio exceeding 40 pg/mL”. The ratio is a dimensionless quantity.

Response:

            The sentence was rewritten and the expression ‘’pg/mL’’ was removed from the text.

            In the context of these valuable suggestions, statistical analysis, findings and discussion sections were rearranged. The points that were intended to be emphasized were specified. New references were added and rearranged in line with your suggestions. The ‘’Discussion’’ and ‘’Introduction’’ sections have been substantially rewritten. Edits made within the text are shown in bold and underlined. English editing was done to the all parts of the article.

            Thank you for your valuable suggestions that made our study more scientific.

Round 2

Reviewer 1 Report

Comments and Suggestions for Authors

All issues are resolved, and the authors responded well